# DRIFT: Directional Reasoning Injection for Fine-Tuning MLLMs

## Abstract

Multimodal large language models (MLLMs) are rapidly advancing, yet their reasoning ability often lags behind that of strong text-only counterparts. Existing methods to bridge this gap rely on supervised fine-tuning over large-scale multimodal reasoning data or reinforcement learning, both of which are resource-intensive. A promising alternative is *model merging*, which interpolates parameters between reasoning-enhanced LLMs and multimodal variants. However, our analysis shows that naive merging is not always a "free lunch": its effectiveness varies drastically across model families, with some (e.g., LLaVA, Idefics) benefiting while others (e.g., Qwen) suffer performance degradation. To address this, we propose Directional Reasoning Injection for Fine-Tuning (DRIFT) MLLMs, a lightweight method that transfers reasoning knowledge in the gradient space, without destabilizing multimodal alignment. DRIFT precomputes a reasoning prior as the parameter-space difference between reasoning and multimodal variants, then uses it to bias gradients during multimodal fine-tuning. This approach preserves the simplicity of standard supervised fine-tuning pipelines while enabling efficient reasoning transfer. Extensive experiments on multimodal reasoning benchmarks, including MathVista and MathVerse, demonstrate that DRIFT consistently improves reasoning performance over naive merging and supervised fine-tuning, while matching or surpassing training-heavy methods at a fraction of the cost.

## 1 Introduction

Multimodal large language models (MLLMs) (Bai et al., 2025; Team et al., 2023; Li et al., 2024b) have recently achieved impressive progress in perception and alignment, enabling them to answer questions about images, analyze charts, and engage in grounded dialogue. However, despite these advances, their reasoning ability remains substantially weaker than that of text-only large language models (LLMs). Across benchmarks in mathematical reasoning (Pan Lu et al., 2024), logical inference (Xiao et al., 2024), and multi-hop question answering (Xiang Yue et al., 2025), a persistent gap emerges: MLLMs can perceive correctly but struggle to chain information into coherent reasoning steps. Bridging this gap is essential for applications that demand not only multimodal understanding but also structured, reliable reasoning.

A mainstream approach to improving reasoning in MLLMs is multimodal supervised fine-tuning (SFT) or reinforcement learning (RL) on reasoning-intensive datasets. Yet both are resource-heavy: collecting multimodal CoT-style data is costly, and reinforcement learning adds instability and computational overhead. In contrast, text-only reasoning models (DeepSeek-AI, 2025) are far easier to obtain due to the growing availability of large-scale text-only CoT resources. This naturally raises a research question: *Can we transfer reasoning from text-only experts into MLLMs efficiently?*

A promising direction is parameter-space model merging, where the weights of a reasoning model are interpolated with those of an MLLM (Chen et al., 2025a). While exciting in its simplicity, our experiments reveal that naive merging is fragile (as shown in Sec. 3.2). It often disrupts perception and alignment, and in many cases even reduces reasoning performance. Learning merge coefficients during fine-tuning partly alleviates this issue, but at the cost of huge training overhead and instability.

To address these limitations, we propose DRIFT, *Directional Reasoning Injection for Fine-Tuning*, a lightweight gradient-based method that transfers reasoning knowledge without destabilizing multimodal training. Rather than interpolating weights in parameter space, DRIFT operates in gradient

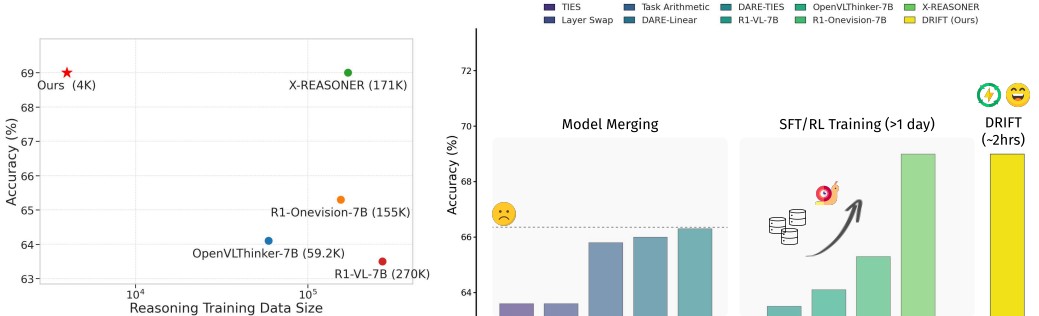

Figure 1: **DRIFT enables efficient reasoning transfer for MLLMs.** *Left:* Compared to reasoning-oriented training methods, DRIFT achieves comparable performance while requiring dramatically less multimodal SFT data (4K vs. >59K examples). *Right:* Simple parameter merging performs poorly on multimodal reasoning benchmarks. Training-based methods improve performance but rely on costly data curation and multi-day training. In contrast, DRIFT reaches competitive results within ~2 hours of training, making it both data- and compute-efficient.

space: it computes a **reasoning vector**, defined as the parameter difference between a reasoning-rich text model and its multimodal counterpart, and uses this as a directional prior to guide updates during multimodal SFT. By injecting this guidance selectively into transformer modules (e.g., attention projections or MLP layers), DRIFT biases optimization toward reasoning while preserving perception. Essentially, DRIFT introduces no additional parameters, requires only a small amount of multimodal reasoning data (as shown in Fig. 1), and integrates seamlessly into existing fine-tuning pipelines.

Our contributions are summarized as follows:

1. We revisit the paradigm of parameter-space model merging for integrating reasoning into MLLMs, showing that while such methods can occasionally yield gains, they are fragile and often degrade performance when models diverge substantially in parameter space.

2. We propose Directional Reasoning Injection for Fine-Tuning (DRIFT), a simple yet effective gradient-based method that leverages the difference between text-only reasoning experts and multimodal models as a directional prior during supervised fine-tuning.

3. Extensive experiments on various multimodal reasoning benchmarks demonstrate that DRIFT consistently outperforms standard SFT and parameter-merging approaches, achieving competitive results with training-heavy methods while requiring less data and compute.

## 2 RELATED WORKS

### 2.1 MULTIMODAL REASONING IN LARGE LANGUAGE MODELS

Following the success of chain-of-thought prompting in enabling large language models (LLMs) to solve complex problems step by step, researchers have increasingly explored whether similar reasoning capabilities exist in multimodal large language models (MLLMs). Among the many domains for evaluation, mathematical reasoning has emerged as one of the most prominent. Lu et al. (2023) introduced MathVista, a visual mathematics benchmark designed to assess the problem-solving abilities of MLLMs on math tasks that require visual understanding. Similarly, Xiao et al. (2024) proposed LogicVista, which evaluates integrated logical reasoning skills over visual concepts. Additional benchmarks, including MathVision (Wang et al., 2024a), MathVerse (Renrui Zhang et al., 2024), and WeMath (Qiao et al., 2024), extend this line of research by covering diverse mathematical problem types and difficulty levels, with a strong emphasis on the vision modality.

Many methods have been proposed to enhance the reasoning ability of MLLMs. Ratzlaff et al. (2025); Li et al. (2024d); Ranaldi & Freitas (2024) explore instruction tuning to teach MLLMs to reason over visual concepts. Similarly, Subramaniam et al. (2025); Huang et al. (2024b); Dong et al. (2025) adopt supervised fine-tuning (SFT) to further improve MLLM performance. More recent works (Wan et al., 2025; Liu et al., 2025b; Chen et al., 2025b) demonstrate that reinforcement learning (RL) approaches

can effectively enhance the reasoning capabilities of MLLMs while maintaining strong generalization across diverse tasks. Among these methods, both SFT and RL have shown remarkable potential. SFT is generally lightweight and efficient, but its effectiveness depends heavily on the availability of high-quality, diverse multimodal datasets. RL methods, on the other hand, are less constrained by dataset diversity and can yield robust improvements, though they are more computationally expensive and require substantial resources for training.

## 2.2 Efficient Fine-Tuning of LLMs

Given the high memory and computational cost of full-parameter fine-tuning, numerous studies have proposed methods to reduce these costs and improve training efficiency. These approaches can generally be divided into parameter-efficient and data-efficient fine-tuning methods.

**Parameter-Efficient Fine-Tuning**. Hu et al. (2022) introduced LoRA, which reduces trainable parameters by injecting and training a low-rank decomposition within the model's weight matrices. Subsequent works have refined LoRA with various enhancements, including QLoRA (Dettmers et al., 2023), LoRA+ (Hayou et al., 2024), and LiSA (Pan et al., 2024). Another line of work focuses on adapter-based methods, where small trainable modules are inserted into the model while keeping the base parameters frozen. Examples include AdaptMLLM (Lankford et al., 2023), LLaMA-Adapter (Zhang et al., 2024b; Gao et al., 2023), and Bt-Adapter (Liu et al., 2024).

**Data-Efficient Fine-Tuning**. Another research direction seeks to improve fine-tuning efficiency by carefully curating or compressing the training data. For instance, Lin et al. (2024) propose pruning and selecting representative samples to maximize data utility. He et al. (2024) leverage external MLLMs to select high-quality multimodal data for training. Additionally, methods such as those proposed by Shang et al. (2024) and Cai et al. (2024) reduce the number of visual tokens used for training, thereby accelerating both fine-tuning and inference.

**Model Merging.** An even more efficient alternative, model merging repurposes fine-tuned models by directly combining parameters through simple arithmetic (Ilharco et al.; Yadav et al., 2023; Yu et al., 2024), requiring no additional training or inference cost. Although well studied in vision models (Huang et al., 2024a; Gargiulo et al., 2025), its use in MLLMs remains limited. Recent work, such as BR2V (Chen et al., 2025a), demonstrates the potential of merging for transferring reasoning into multimodal models. Nonetheless, large parameter discrepancies and cross-modal transfer of reasoning remain open challenges. Our work addresses these by injecting reasoning priors from LLMs into MLLMs via gradient space merging.

## 3 Method

### 3.1 Task Formulation

Starting from a text-only base LLM $\phi$, one can derive multiple variants such as instruction-tuned models or task-specific experts for domains like mathematics, programming, or chemistry. Reasoning can be injected into this base model through two primary approaches: (i) supervised fine-tuning (SFT) on chain-of-thought (CoT) datasets, or (ii) reinforcement learning (RL), incentivizing step-by-step reasoning behavior without explicit CoT labels. To equip the model with visual understanding, a standard strategy is to integrate a visual encoder that maps images into token representations processed jointly with text, then train the encoder and LLM backbone end-to-end.

Despite sharing the same base, reasoning and vision capabilities are often developed in isolation: multimodal large language models rarely inherit the reasoning ability of their text-only counterparts. Building an MLLM capable of reasoning typically requires SFT over costly multimodal CoT data. RL can further refine reasoning, but usually assumes a seed of reasoning ability or sufficient long-context capacity. In contrast, the growing availability of text-only CoT resources makes it often easier to first obtain a strong text-only reasoning model from $\phi$. This imbalance naturally motivates our research question ($\mathcal{Q}$): *can we leverage a text-only reasoning model to guide the transformation of a non-reasoning multimodal LLM into a reasoning-capable one?*

Formally, let the base model be $\phi$ and its variant fine-tuned on a task $T_i$ be denoted $\phi_{T_i}$. Our objective is to efficiently learn a model $\phi_{T'}$ by leveraging $M$ domain experts $\{\phi_{T_1}, \phi_{T_2}, \ldots, \phi_{T_M}\}$, where

Table 1: **Effect of model merging on multimodal reasoning benchmarks.** Performance is reported on MathVista (Pan Lu et al., 2024), MathVision (Ke Wang et al., 2024), and MathVerse (Renrui Zhang et al., 2024) for four multimodal LLMs (LLaVA-Next-8B (Li et al., 2024a), Idefics-8B (Laurençon et al., 2024), Qwen2-VL-7B (Wang et al., 2024b), and Qwen2.5-VL-7B (Bai et al., 2025)) before and after merging with their corresponding text-only reasoning experts.

| Benchmark | LLaVA-Next-LLaMA3-8B | | | Idefics-8B | | | Qwen2-VL-7B | | | Qwen2.5-VL-7B | | |
|---|---|---|---|---|---|---|---|---|---|---|---|---|
| | Base | +Dart-Uniform | *rel.* | Base | +MetaMath | *rel.* | Base | +Qwen2-Math | *rel.* | Base | +DeepSeek-R1 | *rel.* |
| MathVista | 37.4 | 38.2 | **+0.8** | 51.8 | 53.2 | **+1.4** | 61.2 | 60.2 | **-1.0** | 67.9 | 65.8 | **-2.1** |
| MathVision | 13.8 | 15.8 | **+2.0** | 17.1 | 11.8 | **-5.3** | 21.1 | 21.7 | **+0.6** | 25.0 | 22.7 | **-2.3** |
| MathVerse | 16.0 | 17.4 | **+1.4** | 11.0 | 12.4 | **+1.4** | 26.9 | 26.7 | **-0.2** | 41.4 | 33.2 | **-8.2** |

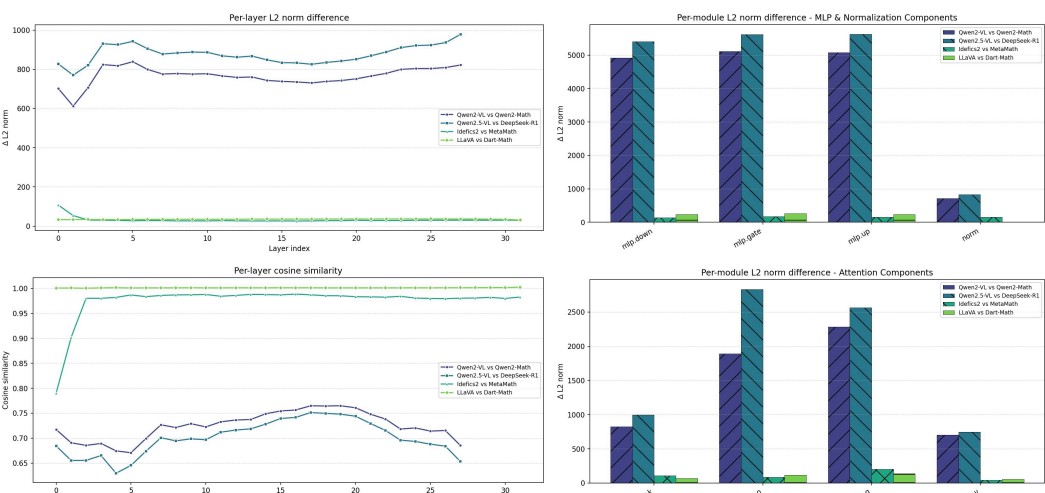

Figure 2: **Layer/Module-wise analysis of model merging pairs.** We compare LLaVA-Next-8B vs. Dart-Uniform, Idefics-8B vs. MetaMath, Qwen2-VL-7B vs. Qwen2-Math-7B, and Qwen2.5-VL-7B vs. DeepSeek-R1-Qwen-7B. *Top Left*: per-layer $\mathcal{L}_2$ norm differences. *Bottom Left*: per-layer cosine similarity. *Top Right*: average $\mathcal{L}_2$ norm differences for FFN layers and normalization layers. *Bottom Right*: average $\mathcal{L}_2$ norm differences for attention projections (Q/K/V/O).

$T' = \{T_1, T_2, \ldots, T_M\}$. In this work, we focus on the case where $T_1 = \underline{\text{text-only reasoning}}$ and $T_2 = \underline{\text{visual understanding}}$, and aim to combine them in a data- and compute-efficient manner to obtain a reasoning-capable multimodal model.

## 3.2 IS MODEL MERGING ALWAYS A "FREE LUNCH"?

Model merging, which combines the weights of domain experts so that the resulting model inherits desirable properties from each, appears to offer a promising path toward our research question. In particular, one can merge a text-only reasoning LLM with the backbone of a multimodal LLM (MLLM) to unify their complementary strengths. Recent work, such as BR2V (Chen et al., 2025a), has explored this direction by attempting to integrate reasoning into multimodal LLM.

To explore the potential of model merging, we apply BR2V to the LLM backbones of a text-only reasoning model and a multimodal LLM, both derived from the same base model. We explore a series of models. Concretely, we experiment with `Mistral-7B` (Jiang et al., 2023), `LLaMA3-8B`, `Qwen-2-7B` (Yang et al., 2024), and `Qwen-2.5-7B` (Bai et al., 2025) as base models; `Dart-Uniform` (Tong et al., 2024), `Meta-Math` (Yu et al., 2023), `Qwen2-Math-7B` (Yang et al., 2024), and `DeepSeek-R1-Distill-Qwen-7B` (DeepSeek-AI, 2025) as text-only reasoning experts; and `LLaVA-Next-LLaMA3-8B` (Li et al., 2024a), `Idefics-8B` (Laurençon et al., 2024), `Qwen2-VL-7B-Instruct` (Wang et al., 2024b), and `Qwen-2.5-VL-7B-Instruct` (Bai et al., 2025) as multimodal variants.

We evaluate the merged models on multimodal reasoning benchmarks, including MathVista (Pan Lu et al., 2024), MathVision (Ke Wang et al., 2024), and MathVerse (Renrui Zhang et al., 2024)

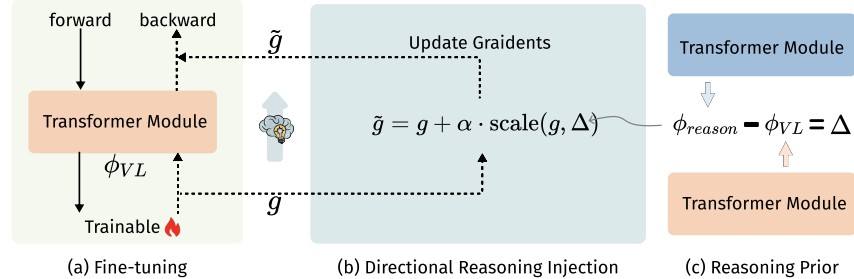

Figure 3: **Overview of Directional Reasoning Injection (DRIFT).** (a) Standard fine-tuning of a multimodal LLM $\phi_{VL}$, where gradients $g$ are applied directly to update trainable modules. (b) DRIFT modifies gradients by injecting a reasoning prior: $\tilde{g} = g + \alpha \cdot \text{scale}(g, \Delta)$, where $\Delta$ encodes the reasoning direction and $\text{scale}(\cdot)$ adjusts how $\Delta$ interacts with $g$. (c) The reasoning prior $\Delta$ is constructed as the parameter difference between a text-only reasoning model $\phi_{\text{reason}}$ and the multimodal variant $\phi_{VL}$. Our method enables reasoning knowledge to be transferred without destabilizing parameter-space merging.

Vision-Only subset (see Tab. 1). While BR2V enhances the reasoning ability of `LLaVA-Next` and `Idefics`, yielding up to a $2\%$ improvement when merged with reasoning-augmented variants, it often causes performance degradation in the `Qwen` series across most test cases.

To further investigate these mismatched behaviors across different models, we compute layer-wise $\mathcal{L}_2$ norm and cosine similarity between model backbones, quantifying both magnitude and directional shifts in parameter space. This analysis enables us to examine how reasoning and visual understanding are distributed in parameter space, thereby characterizing the relationships between post-trained variants derived from the same base LLM.

As shown in Fig. 2, variants of `LLaMA` and `Mistral` remain relatively close in parameter space, while `Qwen` variants are substantially more dispersed. Moreover, the parameter magnitudes of multimodal `Qwen` models diverge sharply from their reasoning counterparts, which likely explains the failure of naive merging in this family. These results suggest that model merging is not universally a "free lunch", its success depends strongly on how post-training reshapes the underlying parameter space.

### 3.3 DIRECTIONAL REASONING INJECTION FOR FINE-TUNING MLLMS

We reformulate the task as mapping a reasoning expert $\phi_{\text{reason}}$ and a multimodal LLM $\phi_{\text{VL}}$ into a reasoning-capable multimodal model:

$$(\phi_{\text{VL}}, \phi_{\text{reason}}) \;\mapsto\; \phi_{\text{VL}\oplus\text{reason}}.$$

As demonstrated in Sec. 3.2, typical merging methods like BR2V (Chen et al., 2025a) merge parameters (task vectors) relative to the base model:

$$\phi_{\text{VL}\oplus\text{reason}} = \phi_{\text{base}} + \alpha(\phi_{\text{VL}} - \phi_{\text{base}}) + (1-\alpha)(\phi_{\text{reason}} - \phi_{\text{base}}). \tag{1}$$

However, this approach often fails in practice. Large discrepancies between $\phi_{\text{VL}}$ and $\phi_{\text{reason}}$ make performance highly sensitive to $\alpha$: even small distributional mismatches can yield large shifts in weights. Learning an optimal $\alpha$ is expensive because it requires storing all candidate models in GPU memory. Moreover, when the two models diverge heavily in magnitude, naive interpolation can cause unstable updates or gradient explosions. These drawbacks suggest that parameter-space merging is neither stable nor efficient for large-scale MLLMs.

**From parameter merging to directional injection.** Instead of interpolating parameters, we propose to inject reasoning knowledge into the *optimization trajectory*. Our key insight is that the gap between variants encodes domain-specific knowledge (e.g., reasoning). Rather than directly applying this gap in weight space, which may distort multimodal alignment, we leverage it as a *directional prior* to guide gradient updates.

We define the difference between a reasoning model and a multimodal variant:

$$\Delta = \phi_{\text{reason}} - \phi_{\text{VL}}, \tag{2}$$

restricted to reasoning-relevant modules (MLP projections, attention projection layers, and normalization layers). This $\Delta$ serves as the *reasoning direction*. During multimodal supervised fine-tuning (SFT) with limited multimodal CoT data, we leave model weights intact and instead bias gradients towards the reasoning direction. For a parameter $w$ with gradient $g$, we compute the guided gradient:

$$\tilde{g} = g + \alpha \cdot \text{scale}(g, \Delta), \tag{3}$$

where $\alpha$ controls prior strength and $\text{scale}(\cdot)$ adjusts how $\Delta$ interacts with $g$. We explore three variants:

- **Absolute:** $\tilde{g} = g + \alpha\Delta$, directly pulling weights toward the reasoning prior.
- **Grad-Norm:** $\tilde{g} = g + \alpha\|g\|\frac{\Delta}{\|\Delta\|}$, aligning updates with the direction of $\Delta$ while preserving the gradient magnitude of $g$.
- **Grad-Norm w/ Adaptive $\alpha$:** $\tilde{g} = g + \alpha'\|g\|\frac{\Delta}{\|\Delta\|}$, where $\alpha' = \alpha \cdot \frac{1+\cos(g,\Delta)}{2}$, adapting strength based on gradient-delta alignment.

**Discussion.** The proposed *Directional Reasoning Injection* (DRIFT) offers two main benefits. First, it preserves the standard multimodal SFT pipeline: training remains on multimodal data, but optimization is nudged toward reasoning directions, enabling gradual knowledge transfer without destabilizing pre-merge operations or requiring large-scale multimodal CoT supervision. Second, it is lightweight: the reasoning prior $\Delta$ is computed once, stored on the CPU, and only transferred to the GPU when needed for gradient updates. DRIFT introduces no additional parameters and modifies only the backward pass, making it both memory-efficient and easily scalable to large MLLMs.

## 4 EXPERIMENTS

### 4.1 DATASET COLLECTION

To enable reasoning transfer, we require multimodal reasoning data, but only in small amounts. Prior work, ThinkLite (Wang et al., 2025), demonstrates that high-quality and challenging questions are more effective for training than larger volumes of easier ones. Building on this insight, we start from the ThinkLiteVL-11K dataset, which contains 11K high-quality image–question pairs. However, this dataset provides only answers without accompanying reasoning chains. To address this, we employ the ThinkLite models (trained on the same data) to distill chain-of-thought (CoT) annotations. We then filter out examples where the model either produces incorrect answers or outputs an invalid format. The retained reasoning traces are enclosed within `<think></think>` tags to clearly separate the chain-of-thought from the final answer. After filtering, we obtain a curated set of 4K high-quality multimodal reasoning examples, which serve as the foundation for our proposed *Directional Reasoning Injection*.

### 4.2 EXPERIMENTAL SETTING

In particular, to construct a strong multimodal reasoning model, we select `DeepSeek-R1-Qwen-Distill-7B` (DeepSeek-AI, 2025) as the text-only reasoning expert and `Qwen2.5-VL-7B-Instruct` (Bai et al., 2025) as the multimodal backbone. The `DeepSeek-R1` family is designed to elicit explicit reasoning traces, while `Qwen2.5-VL` provides strong visual grounding and perception. Investigating whether combining these complementary capabilities yields a more powerful multimodal reasoning model is our central question.

We implement our method on top of the `LLaMAFactory` codebase (Zheng et al., 2024), ensuring reproducibility and compatibility with existing fine-tuning workflows. Training follows the standard supervised fine-tuning pipeline, with DRIFT integrated as a lightweight plug-in. The reasoning direction $\Delta$ is precomputed once and cached on the CPU, then transferred to the GPU only when needed for gradient updates. During backpropagation, we register additional gradient hooks that inject $\Delta$ into online gradients, enabling reasoning-aware optimization with negligible overhead. We train the model for three epochs with a learning rate of $1 \times 10^{-6}$.

For evaluation, we focus on multimodal reasoning benchmarks, particularly those involving mathematical reasoning: MathVista (Pan Lu et al., 2024) testmini subset, MathVision (Ke Wang et al., 2024), MathVerse (Renrui Zhang et al., 2024) vision-only subset, WeMath (Runqi Qiao et al., 2024),

Table 2: **Evaluation results on multimodal reasoning benchmarks.** We compare our gradient-based merging approach with standard parameter-space merging baselines. Results are reported on *MathVista*, *MathVision*, *MathVerse*, *WeMath* (strict/loose), and *LogicVista*. Best results are in **bold**. Note: Improvements are reported relative to Baseline.

| Model | MathVista | MathVision | MathVerse | WeMath strict | WeMath loose | LogicVista | Avg. |
|---|---|---|---|---|---|---|---|
| Qwen2.5-VL-7B-Instruct (Bai et al., 2025) | 67.9 | 25.0 | 41.4 | 34.3 | 52.8 | 46.7 | 44.7 |
| *Parameter merging with DeepSeekR1-Qwen-Distill-7B* | | | | | | | |
| Task Arithmetic (Ilharco et al.) | $65.8_{-2.1}$ | $22.7_{-2.3}$ | $33.2_{-8.2}$ | $30.1_{-4.2}$ | $51.2_{-1.6}$ | $42.0_{-4.7}$ | $40.8_{-3.9}$ |
| Layer Swap (Bandarkar et al.) | $63.6_{-4.3}$ | $22.9_{-2.1}$ | $37.9_{-3.5}$ | $32.1_{-2.2}$ | $50.1_{-2.7}$ | $35.1_{-11.6}$ | $40.3_{-4.4}$ |
| TIES (Yadav et al., 2023) | $63.6_{-4.3}$ | $23.1_{-1.9}$ | $39.5_{-1.9}$ | $33.4_{-0.9}$ | $51.7_{-1.1}$ | $42.1_{-4.6}$ | $42.2_{-2.5}$ |
| DARE-TIES (Yu et al., 2024) | $66.3_{-1.6}$ | $23.6_{-1.4}$ | $38.3_{-3.1}$ | $33.7_{-0.6}$ | $52.6_{-0.2}$ | $42.0_{-4.7}$ | $42.8_{-1.9}$ |
| DARE-Linear (Yu et al., 2024) | $66.0_{-1.9}$ | $22.3_{-2.7}$ | $35.5_{-5.9}$ | $30.8_{-3.5}$ | $51.2_{-1.6}$ | $42.5_{-4.2}$ | $41.4_{-3.3}$ |
| *Reasoning Injection from DeepSeekR1-Qwen-Distill-7B* | | | | | | | |
| DRIFT (Ours) | $\mathbf{69.0}_{+1.1}$ | $\mathbf{26.5}_{+1.5}$ | $\mathbf{44.4}_{+3.0}$ | $\mathbf{36.3}_{+2.0}$ | $\mathbf{58.2}_{+5.4}$ | $\mathbf{45.6}_{-1.1}$ | $\mathbf{50.7}_{+6.0}$ |

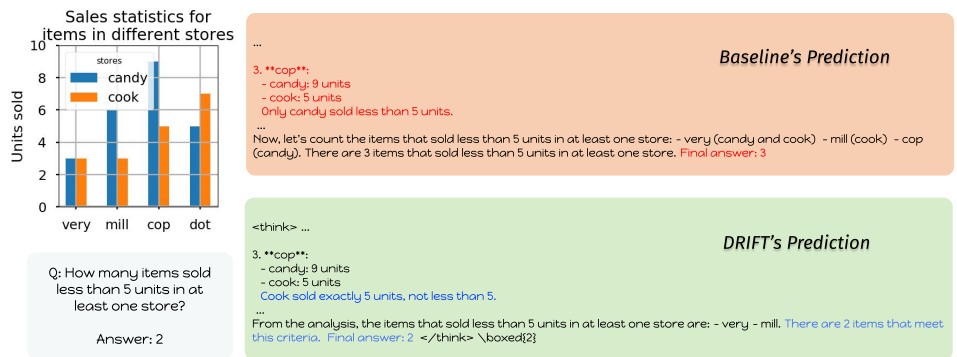

Figure 4: **Qualitative example.** DRIFT corrects a failure mode where the model's visual perception is accurate but the reasoning chain leads to an incorrect answer.

and LogicVista (Xiao et al., 2024). These datasets contain not only general visual question answering tasks but also problems that explicitly require reasoning, making them suitable testbeds for our approach. We adopt `VLMEvalKit` (Duan et al., 2024) for standardized evaluation and to minimize randomness, following the official protocols of each benchmark.

### 4.3 COMPARISON WITH PARAMETER MERGING-BASED METHODS

As discussed in Sec. 3.2, parameter-space merging has emerged as a popular approach for injecting reasoning into multimodal models. However, its effectiveness is far from guaranteed: naive merging often yields no gain, particularly when the underlying models diverge significantly in parameter space. We compare against several representative merging approaches, including Task Arithmetic (Ilharco et al.), Layer Swap (Bandarkar et al.), TIES (Yadav et al., 2023), and DARE (Yu et al., 2024). These methods operate by directly manipulating model weights via vector addition or interpolation, layer replacement, or sparsity/importance masking, to combine complementary skills without full retraining. We follow the hyperparameter selection practice of Chen et al. (2025a) for fair comparison.

As shown in Tab. 2, we merge the strong reasoning model DeepSeek-R1-Qwen-Distill-7B (DeepSeek-AI, 2025) into Qwen2.5-VL-7B-Instruct (Bai et al., 2025). Surprisingly, none of the merging methods improve performance; in fact, several degrade it. We hypothesize that this failure stems from the large distributional discrepancy between the reasoning model and the multimodal variant, consistent with our earlier analysis in Sec. 3.2. This finding underscores the fragility of parameter-level merging and motivates the need for a more robust alternative.

**Our Gradient-based Alternative.** In contrast, DRIFT sidesteps the instability of direct parameter interpolation by explicitly encoding reasoning directions during supervised fine-tuning. The multi-modal model begins with full vision–language capability inherited from the base, and fine-tuning data naturally couples perception and reasoning. DRIFT leverages this setting by nudging gradients

Table 3: **Evaluation results on visual reasoning benchmarks.** We report performance on MathVista, MathVision, MathVerse, WeMath (strict), and LogicVista across *open-source models*, and *reasoning fine-tuning methods*. [†] indicates results reproduced by ourselves. Our DRIFT results are bold, and improvements relative to our SFT baseline are reported.

| Model | MathVista | MathVision | MathVerse | WeMath | LogicVista |
|---|---|---|---|---|---|
| ***Open-source Models*** | | | | | |
| LLaVA-OneVision-7B (Li et al., 2024c) | 62.6 | 17.6 | 17.6 | 17.7 | 32.0 |
| InternLM-XComposer2.5 (Zhang et al., 2024a) | 64.0 | 17.8 | 16.2 | 14.1 | 34.7 |
| InternVL3-8B (Zhu et al., 2025) | 70.5 | 28.6 | 33.9 | 37.5 | 43.6 |
| InternVL2.5-8B (Chen et al., 2024a) | 64.5 | 17.0 | 22.8 | 23.5 | 36.0 |
| InternVL2-8B (Chen et al., 2024b) | 58.3 | 20.0 | 20.4 | 20.2 | 33.6 |
| QvQ-72B-Preview (Team, 2024) | 70.3 | 34.9 | 48.2 | 39.0 | 58.2 |
| Kimi-VL-16B (Team et al., 2025) | 66.0 | 21.8 | 34.1 | 32.3 | 42.7 |
| Qwen2-VL-7B (Wang et al., 2024b) | 61.6 | 19.2 | 25.4 | 22.3 | 33.3 |
| Qwen2.5-VL-7B (Bai et al., 2025) | 67.9[†] | 25.0[†] | 41.4[†] | 34.3[†] | 46.7[†] |
| ***Reasoning Fine-tuning Methods*** | | | | | |
| R1-Onevision-7B (Yang et al., 2025) | 64.1 | 29.9 | 40.0 | – | 61.8 |
| OpenVLThinker-7B (Deng et al., 2025) | 65.3 | 23.0 | 38.1 | 35.2 | 44.5 |
| R1-VL-7B (Zhang et al., 2025) | 63.5 | 24.7 | 40.0 | – | – |
| X-REASONER (Liu et al., 2025a) | 69.0 | 29.6 | – | – | – |
| Ours (SFT) | 68.7 | 25.1 | 42.0 | 33.3 | 45.6 |
| Ours (DRIFT) | **69.0**$_{+0.3}$ | **26.5**$_{+1.5}$ | **44.4**$_{+2.4}$ | **36.5**$_{+3.2}$ | **45.2**$_{-0.4}$ |

slightly toward the reasoning direction, reinforcing reasoning signals without disrupting multimodal alignment. This design yields consistent improvements across benchmarks, surpassing both the baseline and parameter-merging methods (e.g., $+3.2$ points on MathVista compared to Task Arithmetic). These results highlight that DRIFT provides an effective mechanism for transferring reasoning ability (as shown in Fig. 4), offering robustness where parameter-level merging is brittle.

## 4.4 COMPARISON WITH TRAINING-BASED METHODS

A prominent line of work aims to endow multimodal LLMs with reasoning ability through additional training, typically requiring either large-scale multimodal CoT supervision or specialized fine-tuning strategies such as reinforcement learning. Representative examples include R1-OneVision (Yang et al., 2025), OpenVLThinker (Deng et al., 2025), and X-Reasoner (Liu et al., 2025a), all of which demand curated multimodal reasoning datasets and substantial training budgets. As shown in Tab. 3, these approaches achieve competitive performance, but only at the cost of generating or collecting large-scale CoT traces (see Fig. 1 for performance and dataset size comparison).

In contrast, our method avoids such heavy supervision. By introducing *Directional Reasoning Injection*, we leverage a lightweight reasoning prior distilled from a text-only expert and inject it into multimodal training via gradient guidance. This design preserves the simplicity of standard SFT pipelines while enabling efficient reasoning transfer.

Empirically, DRIFT achieves consistent gains over the SFT baseline on MathVista, MathVision, MathVerse, and WeMath, while maintaining comparable results on LogicVista. Although training-heavy methods such as X-Reasoner or R1-OneVision sometimes achieve higher absolute scores, DRIFT reaches competitive performance with orders of magnitude less reasoning-specific data and training time. The efficiency benefits of DRIFT are summarized in Tab. 6, which compares the training regimes: existing reasoning-focused methods require **days of training** with SFT or RL, while DRIFT requires only SFT-style training and completes in **roughly two hours**.

Overall, these results, together with the efficiency analysis, validate our central claim: reasoning transfer can be achieved not only through resource-intensive multimodal fine-tuning, but also via lightweight gradient-space priors that exploit the gap between text-only reasoning experts and multimodal models.

## 4.5 ANALYSIS OF DRIFT

**Is Reasoning Prior Useful?** Tab. 3 shows that simply applying supervised fine-tuning (SFT) provides a strong baseline, yet adding our reasoning prior through DRIFT consistently improves performance.

Table 4: **Comparison of scaling strategies in DRIFT.** We report performance on *MathVista*, *MathVerse*, and *LogicVista*. Scores are shown with relative improvements (*rel.*) over the SFT baseline. Merging candidates include attention projection layers (ATTN), Feedforward layers (MLP), input normalization and output normalization layers (Norm), and the output language model projection head (LM Head).

| | Scaling Strategy | Merge Candidates | MathVista | | MathVerse | | LogicVista | |
|---|---|---|---|---|---|---|---|---|
| | | | Score | *rel.* | Score | *rel.* | Score | *rel.* |
| SFT | – | – | 68.7 | – | 42.0 | – | 45.6 | – |
| DRIFT | Absolute | | 65.7 | **-3.0** | 39.5 | **-2.5** | 25.9 | **-19.7** |
| | Grad-Norm | {ATTN, MLP} | 69.0 | **+0.3** | 44.4 | **+2.4** | 45.1 | **-0.5** |
| | Grad-Norm w/ Relation | | 70.3 | **+1.6** | 43.6 | **+1.6** | 45.6 | 0.0 |
| | | {ATTN} | 69.0 | **+0.3** | 45.3 | **+3.3** | 46.1 | **+0.5** |
| | Grad-Norm | {MLP} | 69.2 | **+0.5** | 42.7 | **+0.7** | 44.7 | **-0.9** |
| | | {ATTN, MLP, Norm} | 68.6 | **-0.1** | 41.6 | **-0.4** | 45.8 | **+0.2** |
| | | {ATTN, MLP, Norm, LM Head} | 69.2 | **+0.5** | 42.1 | **+0.1** | 47.8 | **+2.2** |

For instance, DRIFT achieves +2.4 points on *MathVerse* and +3.2 on *WeMath*, compared to the SFT baseline. These gains suggest that the reasoning prior extracted from text-only experts is indeed useful in guiding multimodal training, providing complementary reasoning signals beyond what the multimodal instruction data alone can supply. Importantly, the improvements are achieved without relying on costly multimodal CoT annotations.

**On the Role of Merging Candidates.** To understand which components benefit most from reasoning injection, we vary the set of modules to which DRIFT is applied (see Tab. 4). We start from the attention layers, and find that applying DRIFT only to attention layers achieves the strongest performance on *MathVerse* (+3.3), with additional improvements on *LogicVista*. In contrast, restricting to feed-forward layers yields modest or inconsistent gains, and including normalization layers often leads to diminished performance. Extending to the LM head provides mixed results – limited impact on *MathVerse* but noticeable gains on *LogicVista*. These findings suggest that attention modules are the most sensitive to reasoning priors, while over-extending to normalization layers can inject noise rather than useful signals.

**On the Role of Merging Strategies.** Different strategies for incorporating the reasoning prior lead to distinct behaviors. The *Absolute* update rule degrades performance across all benchmarks, likely because it pulls parameters too aggressively toward the reasoning model, disrupting multimodal alignment. In contrast, gradient-based scaling strategies (*Grad-Norm* and *Grad-Norm w/ Adaptive* $\alpha$) yield stable improvements. Notably, *Grad-Norm w/ Adaptive* $\alpha$ achieves the highest MathVista score (70.3, +1.6), showing that adapting the prior based on the gradient–delta relation provides a balanced integration. This highlights that subtle guidance, rather than direct overwriting, is the key to successfully transferring reasoning capabilities.

Overall, these analyses reinforce our central claim: reasoning priors are beneficial, but their utility depends strongly on *where* they are applied (attention layers vs. others) and *how* they are integrated (gradient guidance vs. absolute interpolation). DRIFT's design, which biases gradients rather than parameters, provides a stable mechanism for exploiting these priors.

## 5 CONCLUSION

In this work, we explore transferring reasoning from text-only LLMs to multimodal LLMs without large-scale multimodal CoT supervision. While parameter-space merging can yield occasional gains, it often breaks down when models diverge. To overcome this, we propose *Directional Reasoning Injection for Fine-Tuning* (DRIFT), a gradient-based method that guides MLLM fine-tuning with reasoning priors from expert models. DRIFT achieves consistent improvements over SFT and remains competitive with costly reasoning-specific training, showing that lightweight gradient-space priors provide an efficient and scalable path for cross-domain capability transfer.

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

# A APPENDIX

## A.1 PARAMETER-SPACE MERGING METHOD SETUP

We experiment with several parameter-space merging strategies, where models are combined without additional training by directly manipulating their parameters. The hyper-parameters in Tab. 5 correspond to: (i) $\lambda$ coefficients that control the interpolation ratio between two models (Ilharco et al.); (ii) $\alpha$ scaling factors used in data-aware reweighting (e.g., in DARE (Yu et al., 2024)); and (iii) for layer swapping (Bandarkar et al.), the number of layers replaced.

| Method | Hyper-parameters |
|---|---|
| Baseline | - |
| Task Arithmetic | $(\lambda = 0.9, \, 0.1)$ |
| TIES | $(\lambda = 1.6, \, \alpha = 0.2)$ |
| Dare-TIES | $(\lambda = 1.6, \, \alpha = 0.2)$ |
| Dare-Linear | $(\lambda = 1.6, \, \alpha = 0.2)$ |
| Layer Swap | $(\lambda = 0.9, \, 0.1, \, k = 5)$ |

Table 5: Hyper-parameter setup for different parameter-space merging methods. $(\lambda, \alpha, k)$ denote interpolation ratios, scaling factors, and number of swapped layers, respectively.

## A.2 TRAINING TIME COMPARISON

Training efficiency is a critical factor when scaling reasoning-capable MLLMs. Most existing approaches rely on either large-scale supervised fine-tuning (SFT) with multimodal CoT data or reinforcement learning (RL) on specialized reasoning benchmarks. Both settings typically require multiple days of training on high-end GPU clusters, limiting their practicality for rapid iteration or deployment.

As summarized in Tab. 6, representative methods such as OpenVLThinker, R1-OneVision, and X-REASONER all involve either full SFT or RL and require more than one day of training. In contrast, our method, DRIFT, requires only SFT-style training with gradient guidance and completes within roughly two hours under comparable hardware. This dramatic reduction in cost is achieved because DRIFT (i) avoids a huge amount of multimodal CoT data collection, (ii) adds only lightweight gradient-time operations with a precomputed prior, and (iii) leaves the forward pass unchanged.

| Method | SFT | RL | Est. time |
|---|---|---|---|
| OpenVLThinker-7B (Deng et al., 2025) | ✓ | ✗ | $> 1$ day |
| R1-OneVision-7B (Yang et al., 2025) | ✓ | ✗ | $> 1$ day |
| X-REASONER (Liu et al., 2025a) | ✓ | ✓ | $> 2$ days |
| **Ours (DRIFT)** | ✓ | ✗ | $\approx 2$ hrs |

Table 6: **Training schemes and estimated wall-clock cost.** Existing methods require at least one day of training, while DRIFT completes in about two hours under comparable hardware.

In practice, this efficiency means DRIFT can be integrated into existing SFT pipelines with negligible additional overhead, making it far more scalable for both research and production settings.

## A.3 DATASET COLLECTION DETAILS

We leverage the ThinkLite (Wang et al., 2025) model to distill multimodal reasoning data on the ThinkLite-VL-Hard-11K dataset. The prompt used to elicit reasoning traces is illustrated in Figure 5.

After generating candidate responses, we apply a multi-step filtering process to ensure data quality. First, we verify whether the final answer is enclosed in \boxed{} and matches the ground-truth solution. Second, we check the correctness of the reasoning format enclosed by <think> and

**System Prompt:** You FIRST think about the reasoning process as an internal monologue and then provide the final answer. The reasoning process MUST BE enclosed within <think> </think> tags. The final answer MUST BE put in \boxed{}.

**User Prompt:** <question>

Figure 5: Example prompt used to distill reasoning traces from the ThinkLite model.

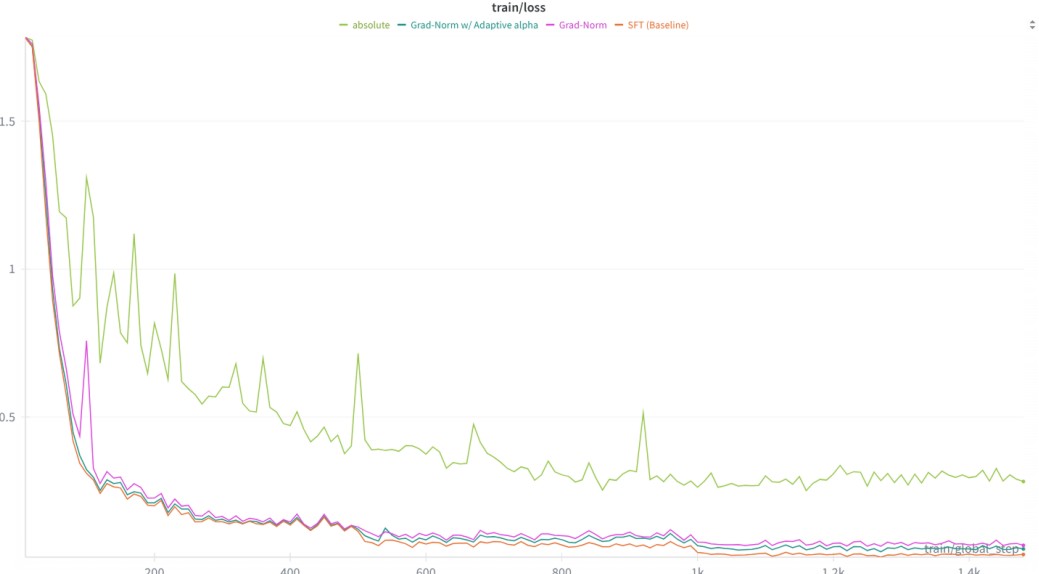

Figure 6: Training loss curves for different gradient merging strategies compared with the SFT baseline. Adaptive Grad-Norm achieves stable optimization while improving performance over standard SFT.

</think>. Finally, we retain the highest-quality subset, resulting in 4K verified samples from the original 11K examples.

## A.4 TRAINING LOSS OF GRADIENT MERGING STRATEGIES

We compare training loss curves of different gradient merging strategies against the SFT baseline on the same dataset. As shown in Fig. 6, the *Absolute* strategy introduces instability, leading to large spikes in the early stages. *Grad-Norm* reduces this effect but still shows noticeable fluctuations. In contrast, *Grad-Norm with Adaptive* $\alpha$ closely follows the stable SFT baseline while yielding improved convergence.

- **Absolute:** $\tilde{g} = g + \alpha\Delta$, directly pulling weights toward the reasoning prior.
- **Grad-Norm:** $\tilde{g} = g + \alpha\|g\|\frac{\Delta}{\|\Delta\|}$, aligning updates with the direction of $\Delta$ while preserving the gradient magnitude of $g$.
- **Grad-Norm w/ Adaptive** $\alpha$**:** $\tilde{g} = g + \alpha'\|g\|\frac{\Delta}{\|\Delta\|}$, where $\alpha' = \alpha \cdot \frac{1+\cos(g,\Delta)}{2}$ adapts the strength based on gradient–delta alignment.

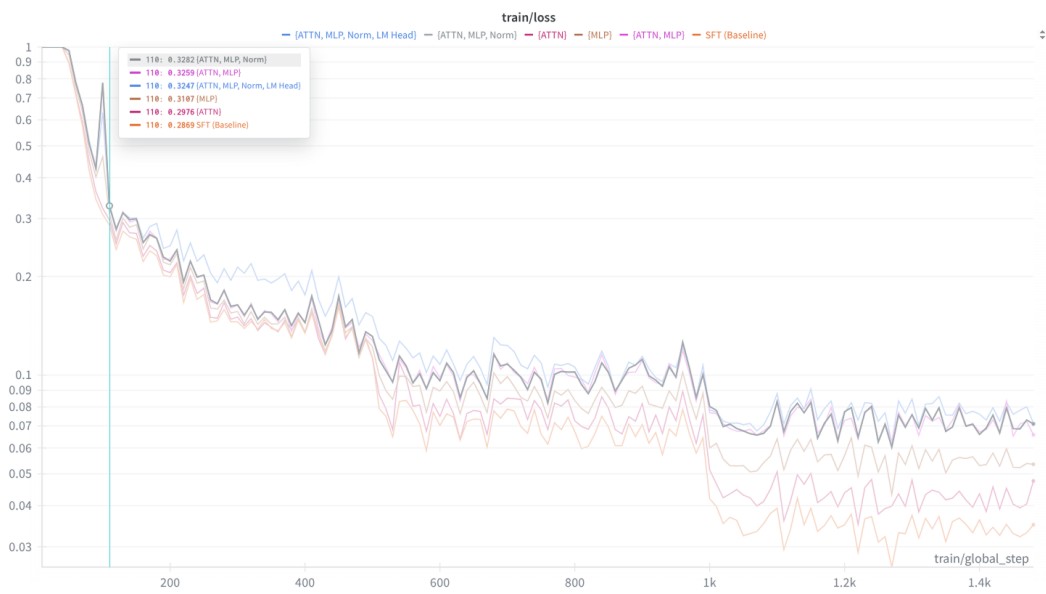

Figure 7: Training loss curves for gradient merging candidates compared with the SFT baseline. The {ATTN} strategy avoids training spikes, while other candidates show instability before convergence.

### A.5 TRAINING LOSS OF GRADIENT MERGING CANDIDATES

We compare the training loss curves of different gradient merging candidates against the SFT baseline on the same dataset. As shown in Fig. 7, merging on {ATTN} yields the most stable curve without spikes, while all other variants exhibit noticeable fluctuations in the early training stage. For clarity, we also plot the loss in log scale and zoom in around the spike region to highlight differences across methods:

- {ATTN}
- {MLP}
- {ATTN + MLP}
- {ATTN + MLP + Norm}
- {ATTN + MLP + Norm + LM Head}

### A.6 LIMITATIONS

While DRIFT demonstrates that lightweight gradient-space priors can effectively transfer reasoning from text-only experts to multimodal models, several limitations remain. First, our method relies on the availability of strong text-only reasoning experts, which constrains applicability in domains where such experts are weak or unavailable. Second, although DRIFT avoids destabilizing multimodal alignment in our experiments, its reliance on precomputed reasoning directions may introduce biases or diminish performance when tasks require perception-heavy reasoning. Third, we primarily evaluate on mathematical and logical reasoning benchmarks; further validation on diverse multimodal tasks such as commonsense reasoning, scientific understanding, or open-domain visual question answering is needed to assess generality. Finally, while DRIFT reduces training costs compared to reinforcement learning or large-scale multimodal CoT supervision, it still adds overhead relative to standard SFT and does not yet guarantee interpretability of the injected reasoning signals.

### A.7 FUTURE WORK

Building on our findings, several directions remain open for exploration. First, extending DRIFT beyond mathematical and logical reasoning to domains such as scientific understanding, embodied

perception, and real-world decision-making would test its generality. Second, developing adaptive strategies that dynamically select or combine reasoning priors, rather than relying on a fixed direction, could improve robustness when transferring across diverse tasks. Third, integrating DRIFT with reinforcement learning or preference optimization may further enhance reasoning without sacrificing multimodal grounding. Finally, improving interpretability of injected reasoning signals, through visualization or attribution, would provide stronger insights into how reasoning knowledge is transferred, fostering trust and transparency in multimodal systems.

### A.8 BROADER IMPACT

This work highlights a lightweight path for transferring reasoning abilities from text-only experts to multimodal models, offering efficiency benefits and reduced reliance on costly multimodal supervision. By lowering the resource barrier, DRIFT may help democratize access to multimodal reasoning systems in academic and industrial settings. However, transferring reasoning across domains also raises important considerations. First, biases embedded in text-only experts may propagate into multimodal models, amplifying inaccuracies or cultural biases in downstream tasks. Second, more capable multimodal reasoning systems may be misused in sensitive domains such as surveillance, misinformation generation, or automated decision-making, where reliability and transparency are critical. Third, although DRIFT reduces compute costs, it still benefits institutions with access to pretrained reasoning experts, potentially reinforcing existing inequities in model development.

## B USE OF LLMS

In this work, large language models were employed exclusively for grammar refinement and language polishing. All substantive contributions—including the design of the conceptual framework, development of algorithms, model training, experimental studies, and the writing of technical content—are entirely original and carried out by the authors.

