# OpenReview forum: "DRIFT: Directional Reasoning Injection for Fine-Tuning MLLMs"
_ICLR.cc/2026/Conference — ICLR 2026 Conference Withdrawn Submission_

### Official Review · Reviewer_QU8a · 2025-10-30

**Soundness:** 2
**Presentation:** 2
**Contribution:** 2
**Rating:** 2
**Confidence:** 4

**Summary:**

This paper addresses the weak reasoning ability of multimodal large language models (MLLMs) and the high cost of existing enhancement methods. It proposes DRIFT, which computes a reasoning prior as the parameter difference between a reasoning-capable LLM and an MLLM, then uses it to guide gradient updates during fine-tuning, effectively injecting reasoning skills while preserving multimodal alignment. Experiments show that DRIFT outperforms standard fine-tuning and naive model merging on benchmarks like MathVista and MathVerse, achieving performance comparable to costly methods at a fraction of the training expense.

**Strengths:**

The proposed method is highly efficient in design; the paper provides extensive experiments to demonstrate its effectiveness.

**Weaknesses:**

On Experimental Results:

1. Some of the reported experimental results differ from those originally published in the paper. For instance, the performance of OpenVLThinker-7B on Mathvista has decreased from 72.3 to 65.3. What could be the reason for this discrepancy? It's crucial to clarify whether this is due to differences in experimental setup, dataset versions, or other factors.

2. Experiments should also cover models of varying sizes, including both larger models (e.g., 32B) and smaller ones, to provide a comprehensive understanding of the method's scalability and performance across different capacities.

On Methodology:

3. Previous works have similarly explored methods based on model gradients for merging. The manuscript should clearly articulate the technical distinctions and innovations of the current approach compared to these prior efforts, emphasizing what makes it unique or advantageous.

On Presentation Details:

4. In Figure 2, the caption font size is too small, making it difficult to read. It is recommended to increase the font size for better readability.

5. In Figure 4, the legend partially obscures the results, making it hard to discern where the advantages lie. Adjustments should be made to ensure all data points and comparisons are clearly visible and understandable.

6. The formatting of Table 3's title appears incorrect, with abnormal spacing on the second line. This should be corrected to adhere to standard formatting guidelines for clarity and professionalism.

[1] EDITING MODELS WITH TASK ARITHMETIC, 2023.03
[2] FAST MODEL EDITING AT SCALE, 2022.06

**Questions:**

See Weakness.

---

### Official Review · Reviewer_a7Fe · 2025-11-01

**Soundness:** 2
**Presentation:** 3
**Contribution:** 3
**Rating:** 4
**Confidence:** 3

**Summary:**

This paper aims to narrow the gap in reasoning performance between MLLMs and their LLM counterparts. A method called DRIFT is proposed which conditions gradient updates during training on the difference in parameter space between the MLLM and the LLM. Experiments are conducted which show that DRIFT achieves better improvements in reasoning performance than alternative methods such as model merging while avoiding degradation in multimodal alignment.

**Strengths:**

1. This work addresses an important and timely problem, which is mitigating the gap in reasoning capabilities seen in MLLMs relative to the LLMs from which they were derived.
2. Conditioning gradient updates on the difference in parameter space between MLLMs and LLMs is an interesting and novel approach (to the best of my knowledge).
3. Several interesting analyses are provided investigating the usefulness of reasoning, the role of merging candidates, and merging strategies.

**Weaknesses:**

1. The discussion of related work contains errors - Ratzlaff et al. (2025) is a training-free model merging technique that merges LLM parameters into an MLLM. L104-106 incorrectly states that it is an instruction tuning technique.
2. The experimental results focus exclusively on math & logical reasoning tasks. A broader range of reasoning tasks here would help in assessing how well the method generalizes.
3. DRIFT seems to offer no or only negligible improvement relative to SFT on 2/5 datasets (MathVista and LogicVista).
4. The authors make the case that model merging methods fail to offer any improvements for MLLMs (Table 2), but it's unclear if any tuning of merging parameters was done for these experiments, which can significantly impact the results.

**Questions:**

1. In collecting the dataset (section 4.1), was any analysis done to verify the accuracy of the reasoning chains?
2. Did you try tuning merging parameters for any of the evaluated model merging techniques?
3. L259-260: why exactly is it necessary to store all candidate models in GPU memory when trying to optimize the merging strength? Couldn't different models be merged with different parameter values and evaluated separately on a validate set to choose the optimal value?

---

### Official Review · Reviewer_iacC · 2025-11-01

**Soundness:** 2
**Presentation:** 2
**Contribution:** 2
**Rating:** 4
**Confidence:** 4

**Summary:**

This paper introduces DRIFT, a lightweight method to transfer reasoning capabilities from text-only language models to multimodal large language models (MLLMs) without destabilizing their visual alignment. DRIFT computes a "reasoning direction" as the parameter difference between a reasoning-rich text model and a multimodal model, then uses it to bias gradients during fine-tuning. It achieves strong performance on multimodal reasoning benchmarks like MathVista and MathVerse using only ~4K training examples and ~2 hours of training, outperforming other parameter merging and rivaling data-heavy methods.

**Strengths:**

- Instead of merging parameter, DRIFT injects reasoning priors into gradients space.
- Clearly motivates the problem, explains the fragility of naive merging, and presents DRIFT as a simple yet effective alternative.

**Weaknesses:**

- Limited generalization: Only tested on Qwen2.5-VL-7B; needs validation on LLaVA, Idefics, InternVL, and larger scales (14B–72B).
- Interpretability of Δ: No evidence that parameter difference encodes reasoning, not just task-specific drift.
- Avg. gain mismatch: Table 2 claims +6.0 mean vs. max +5.4 per benchmark—please clarify.
- Transfer efficiency: DRIFT gains are modest vs. large lift from DeepSeek-R1-Qwen-Distill-7B on text model; how to evaluate the efficiency of transferring reasoning ability?

**Questions:**

- Can Δ be adapted during training? Have you tried dynamic priors (e.g., updating Δ via exponential moving average) to handle evolving reasoning needs?
- Scalability to larger models? Any preliminary results on 14B/32B/72B models? Does parameter divergence grow linearly, or does DRIFT need adjustment?
- Interpretability of injected reasoning? Why the difference between text-only reasoning model and mllms indicates the priors to guide optimization? For example, can you visualize which attention heads are most influenced by Δ?

---

### Official Review · Reviewer_9SMS · 2025-11-03

**Soundness:** 3
**Presentation:** 3
**Contribution:** 3
**Rating:** 4
**Confidence:** 3

**Summary:**

This paper tackles an important problem: MLLMs typically lag behind pure text-based LLMs in reasoning ability. Existing approaches based on SFT or RL are computationally expensive, while seemingly cheaper alternatives such as model merging are highly unstable (as convincingly demonstrated in Section 3.2, especially on the Qwen models).

To address this, the authors propose DRIFT (Directional Reasoning Injection for Fine-Tuning), a lightweight gradient-space injection method. DRIFT precomputes a reasoning prior vector ($\Delta = \phi_{reason} - \phi_{VL}$), and during SFT, biases the gradient updates using this vector ($\tilde{g} = g + \alpha \cdot \text{scale}(g, \Delta)$). The authors claim that this approach is both data- and compute-efficient (requiring only 4K samples and about two hours of training), while achieving state-of-the-art or competitive results on mathematical reasoning benchmarks such as MathVista.

**Strengths:**

1. The paper has a clear motivation and addresses a key problem in the MLLM domain. It analyzes the effect of naive model merging on enhancing MLLM reasoning ability and demonstrates the fragility of such merging methods.
2. The proposed DRIFT method is innovative: instead of performing unstable parameter-space merging, it guides learning in the gradient space, providing a gentler way to inject reasoning capability. The method is also efficient in terms of both data.
3. DRIFT outperforms naive merging methods on reasoning benchmarks, and the model trained with ThinkLite surpasses its SFT counterpart on mathematical reasoning tasks.

**Weaknesses:**

1. The core motivation of the paper is to demonstrate that model merging fails on the Qwen series but works effectively on LLaMA- and Mistral-based models. However, in the main experiments, the authors only conduct studies based on Qwen models. The authors should at least include experiments on one model family where merging is effective to validate the generality of DRIFT.
2. The paper evaluates models only on mathematical reasoning tasks. While math is a representative form of reasoning, reasoning is not limited to mathematics. It is recommended to include evaluations on non-mathematical visual reasoning tasks.
3. The authors claim that DRIFT “transfers reasoning knowledge without destabilizing multimodal alignment”, but this claim lacks direct evidence, as all evaluations are conducted on mathematical tasks. It remains unclear whether the general multimodal perception ability is destabilized.
4. The training of DRIFT is based on the high-quality ThinkLite dataset, and models trained with ThinkLite alone already achieve near SOTA performance. DRIFT provides only about a 2 point average improvement on top of this, which somewhat weakens its overall contribution.

**Questions:**

See weaknesses

---

### Note · Authors · 2026-01-06

I have read and agree with the venue's withdrawal policy on behalf of myself and my co-authors.